# *Enterococcus faecalis* Endophthalmitis: Clinical Settings, Antibiotic Susceptibility, and Management Outcomes

**DOI:** 10.3390/microorganisms9050918

**Published:** 2021-04-24

**Authors:** Kuan-Jen Chen, Chi-Chun Lai, Hung-Chi Chen, Ying-Jiun Chong, Ming-Hui Sun, Yen-Po Chen, Nan-Kai Wang, Yih-Shiou Hwang, An-Ning Chao, Wei-Chi Wu, Ling Yeung, Chi-Chin Sun, Laura Liu, Yi-Hsing Chen, Hung-Da Chou

**Affiliations:** 1Department of Ophthalmology, Chang Gung Memorial Hospital, Taoyuan 333, Taiwan; chichun.lai@gmail.com (C.-C.L.); chen.agi@gmail.com (H.-C.C.); chongsm@hotmail.com (Y.-J.C.); minghui0215@gmail.com (M.-H.S.); yenpo.chen@gmail.com (Y.-P.C.); yihshiou.hwang@gmail.com (Y.-S.H.); anningchao@hotmail.com (A.-N.C.); weichi666@gmail.com (W.-C.W.); arvin.sun@msa.hinet.net (C.-C.S.); lauraliu@gmail.com (L.L.); yihsing@gmail.com (Y.-H.C.); hdmorph@gmail.com (H.-D.C.); 2College of Medicine, Chang Gung University, Taoyuan 333, Taiwan; lingyeung@gmail.com; 3Department of Ophthalmology, Chang Gung Memorial Hospital, Keelung 204, Taiwan; 4Department of Ophthalmology, Penang General Hospital, Pulau Pinang 10990, Malaysia; 5Department of Ophthalmology, Tucheng Municipal Hospital, Tucheng, New Taipei 236, Taiwan; 6Department of Ophthalmology, Edward S. Harkness Eye Institute, Columbia University, New York, NY 10032, USA; wang.nankai@gmail.com

**Keywords:** antibiotic susceptibility, endophthalmitis, *Enterococcus faecalis*, pars plana vitrectomy, vitreous tap

## Abstract

*Enterococcus faecalis* is known to cause severe acute endophthalmitis and often leads to poor visual outcomes in most ophthalmic infections. This retrospective study is to report the clinical settings, antimicrobial susceptibility patterns, and visual outcome of *E. faecalis* endophthalmitis at a tertiary referral institution in Taoyuan, Taiwan. *E. faecalis* endophthalmitis was diagnosed in 37 eyes of 37 patients. Post-cataract surgery was the most common cause (*n* = 27, 73%), followed by bleb-associated (*n* = 3, 8%), endogenous (*n* = 2, 5%), corneal ulcer-related (*n* = 2, 5%), post-vitrectomy (*n* = 1, 3%), post-pterygium excision (*n* = 1, 3%), and trauma (*n* = 1, 3%). Visual acuities upon presentation ranged from counting fingers to no light perception. Pars plana vitrectomy with intravitreal antibiotics were performed in 23 eyes (76%) as primary or secondary treatment. All isolates (37/37, 100%) were sensitive to vancomycin, penicillin, ampicillin, and teicoplanin. Six of 22 eyes (27%) were resistant to high-level gentamicin (minimum inhibitory concentration > 500 mg/L). Final visual acuities were better than 20/400 in 11 eyes (30%), 5/200 to hand motions in 4 eyes (11%), and light perception to no light perception in 22 eyes (59%). Three eyes were treated with evisceration. Compared with non-cataract subgroups, the post-cataract subgroup showed a significant difference of better visual prognosis (*p* = 0.016).

## 1. Introduction

*Enterococcus faecalis* is a Gram-positive bacterium found in pairs and chains which is part of the normal flora in the human gastrointestinal tract. *E. faecalis* infections are more likely in elderly and debilitated patients, patients with disruption of epithelial or mucosal barriers, and in patients with altered normal flora due to antibiotic treatments. Commensal bacteria that form the normal flora of the conjunctival sac are mainly *Staphylococcus* and *Corynebacterium* species in healthy subjects [1]. *E. faecalis* is detected more frequently in the conjunctival sac of elderly healthy subjects [1]. Thus, these healthy subjects have a higher risk of developing post-operative bacterial endophthalmitis when there is a dissatisfactory peri-operative disinfection or disruption of ocular surface epithelial barriers. *E. faecalis*, a relatively rare cause of endophthalmitis, is identified as a pathogen in both exogenous and endogenous etiologies. Exogenous *E. faecalis* endophthalmitis is primarily caused by trauma and post-surgery, such as cataract extraction, trabeculectomy, penetrating keratoplasty, vitrectomy, and intravitreal injection [2,3,4,5,6,7,8,9,10,11,12,13,14,15,16,17,18]. *E. faecalis* accounted for approximately 1% of culture-positive acute post-cataract surgery endophthalmitis cases in the Endophthalmitis Vitrectomy Study [5,6]. *E. faecalis* is known to cause severe acute endophthalmitis and often leads to poor visual outcomes in most ophthalmic infections [7,8,9,10,11,12,13,14,15,16,17,18].

A previous study reported the antibiotic susceptibility and management outcomes for post-cataract endophthalmitis caused by *E. faecalis* from the years 1996 to 2007 [9]. The aim of the present study, which focuses on the same tertiary institution as described in the previous study [9], is to provide an update on the clinical profile, antibiotic susceptibility, and visual outcomes in another subsequent 12-year (2008–2019) consecutive case series of culture-proven endophthalmitis caused by *E. faecalis*. To the best of our knowledge and at the time of publishing, this study is the largest consecutive case series on *E. faecalis* endophthalmitis.

## 2. Methods and Materials

The Institutional Review Board of Chang Gung Memorial Hospital in Taoyuan, Taiwan, approved this retrospective study protocol (201900614B0C601, 10 August 2019) and waived the need for written informed consent from the study patients. All clinical procedures were conducted according to the principles of the Declaration of Helsinki. The study period began after an electronic medical record system was instituted at Chang Gung Memorial Hospital, Taoyuan, Taiwan, and histories of all patient with *E. faecalis* endophthalmitis were available for review from January 2008 through April 2019. Patient demographics, past ocular and systemic history, presenting ocular symptoms, interval between event and diagnosis of endophthalmitis, ocular examination, culture results, antibiotic susceptibility and resistance patterns, treatment, secondary ocular sequelae, final visual acuity (VA), and duration of follow-up, were all evaluated. All microbiology investigations were performed at the Department of Microbiology, Chang Gung Memorial Hospital, Taoyuan, Taiwan [9]. Bacterial culture isolates were identified by conventional microbiological methods [9] and matrix-assisted laser desorption/ionization time of flight mass spectrometry (MALDI-TOF-MS, Bruker Daltonics, Bremen, Germany). Conventional microbiological methods included Gram-staining and biochemical tests. In MALDI-TOF-MS, automatic measurement of the spectrum and comparative analysis with reference spectra of bacteria were performed using an Ultraflextreme mass spectrometer and MALDI-Biotyper 3.0 software (Bruker Daltonics). The reliability of identification in the MALDI Biotyper system was expressed in points. A log(score) ≥2.0 indicated identification to the species level. The isolates were tested for susceptibility to various antibiotics using the Kirby Bauer Disc diffusion method on Mueller Hinton blood agar. The Clinical and Laboratory Standards Institute (Wayne, PA, USA) standards were used for interpretation and quality control for each corresponding year [19]. Because this was a retrospective study, the antibiotic sensitivity tests for *E. faecalis* isolates were only performed in some antibiotics, including penicillin, ampicillin, vancomycin, teicoplanin, and high-level gentamicin (minimum inhibitory concentration >500 mg/L). The treatment strategies were determined by the respective attending consultant ophthalmologists and did not follow a standardized protocol [9]. Before *S. pneumoniae* was cultured, intravitreal antibiotics included vancomycin with either ceftazidime or amikacin. After positive cultures of *S. pneumoniae* isolates and antibiotic susceptibility testing results were obtained, intravitreal vancomycin was administered. The doses of intravitreal antibiotics were as follow: vancomycin (1 mg/0.1 mL), ceftazidime (2.25 mg/0.1 mL), amikacin (0.2–0.4 mg/0.1 mL), and dexamethasone (0.4 mg/0.1 mL).

For statistical analysis, Snellen visual acuities were converted to the logarithm of the minimal angle of resolution (LogMAR) scale. Approximations for VA worse than 20/2000 were as follow: counting fingers (CF), 20/2000; hand motions (HM), 20/4000; light perception (LP), 20/8000; and no light perception (NLP), 20/16000. For LogMAR values, the following previously used scale was applied: CF, 2.00; HM, 2.30; LP, 2.60; NLP, 2.90. Poor visual outcomes were defined as VA worse than 20/400, whereas favorable prognosis was defined by VA of 20/400 or better. Statistical analysis was performed using SPSS for Windows, version 23 (SPSS Science, Chicago, IL, USA).

## 3. Results

During the 12-year period, 457 isolates were cultured from 390 patients with cultured-proven endophthalmitis. Thirty-seven (8.1%) eyes of 37 patients (9.5%) with *E. faecalis* endophthalmitis were seen during the study period. Table 1 shows demographic, systemic illness, clinical features, management, and visual outcomes of patients with *E. faecalis* endophthalmitis. The median age was 73.0 ± 7.9 years (range, 49 to 90 years). The median follow-up interval was 28 months (range, 1 to 97 months). Significant systemic medical illnesses included primary hypertension in 13 patients and diabetes mellitus in 10 patients.

### 3.1. Patient Clinical Settings and Features

The clinical settings include 27 cases which arose following post-cataract surgery (73%), 3 that were trabeculectomy-associated (8%), 2 that were endogenous (5%), 2 that were corneal ulcer-related (5%), 1 arising through post-vitrectomy (3%), 1 post-pterygium excision (3%), and 1 trauma-related (3%) case. Among the 2 eyes with infectious corneal ulcer-related endophthalmitis, one had history of neurotrophic corneal ulcer, whilst the other arose after a history of traumatic corneal perforation one year previously. The single case of post-pterygium excision developed corneoscleral ulcer-related endophthalmitis 2 months after the pterygium excision procedure. The 2 cases of endogenous *E. faecalis* endophthalmitis were associated with immunocompromised patients with no history of ocular surgery or trauma. However, the primary site of infection where the organism disseminated in the setting of septicemia was not identified. All of the post-cataract patients had acute-onset *E. faecalis* endophthalmitis (range, 1 to 13 days; median, 3 days). The 3 trabeculectomy-associated cases developed endophthalmitis after 1 year, 3 years, and 5 years post-trabeculectomy, respectively.

The presenting symptoms were eye pain in 35 of 37 eyes (92%) and reduced VA in all 37 of 37 eyes (100%), whilst the mean duration of the two symptoms were 2.0 and 2.4 days respectively. The presenting VA was LP in 17 (46%) of 37 patients, HM in 11 (30%) patients, NLP in 8 (22%) patients, and CF in 1 (3%) patient. The mean logMAR VA in this study at presentation was 2.567 ± 0.235 (about light perception). Hypopyon was shown in 35 of 37 eyes (94%), and ocular hypertension (intraocular pressure >21 mm Hg) was present in 21 of 37 eyes (57%). The fundal view was completely non-visualized in all 37 patients due to severe anterior segment inflammation and media opacities.

### 3.2. Microbiology and Antibiotic Susceptibility Testing

*E. faecalis* was identified in vitreous samples in 35 (95%) of 37 patients and in the anterior chamber fluid samples of 19 (51%) patients. All cultures except one were monomicrobial. The culture of one trauma-related *E. faecalis* endophthalmitis yielded polymicrobial growth of *Klebsiella pneumoniae*, *E. faecalis, Enterobacter cloacae*, *Aeromonas hydrophilia*, *Bacillus*, and *E. avium.* The antibiotic susceptibility testing of *E. faecalis* showed that all isolates (37/37, 100%) were susceptible to vancomycin, penicillin, ampicillin, and teicoplanin. Six of 22 (27%) were resistant to high-level gentamicin (minimum inhibitory concentration >500 mg/L). In the post-cataract endophthalmitis group, five of 14 (36%) were resistant to high-level gentamicin.

### 3.3. Management

In the primary management of all the *E. faecalis* endophthalmitis cases, 19 eyes (51%) were given intravitreal antibiotics after vitreous tapping while the other 18 eyes (49%) were treated surgically through pars plana vitrectomy (PPV) followed by immediate post-operative intravitreal antibiotics. Amongst these 2 groups, 16 eyes (43%) were also given intravitreal dexamethasone as part of the primary treatment. Within the first 2 weeks post-primary treatment, 15 (41%) of 37 eyes underwent sessions of vitreous tap and intravitreal antibiotics as part of the follow-up treatment due to clinical evidence of worsening inflammation and infection. On the other hand, another 5 patients underwent PPV with intravitreal antibiotics. From the 18 patients who were initially treated with PPV and intravitreal antibiotics, 6 patients did not need further surgical intervention or intravitreal injections. One patient, who displayed complications with recurrent endophthalmitis 45 days after the initial infection, managed to achieve a final vision of 20/50 after being successfully treated with vitreous tapping and intravenous antibiotics.

### 3.4. Final Visual Outcomes

Final visual acuities were 20/400 or better in 11 of 37 eyes (30%), 5/200 to HM in 4 eyes (11%), and LP to NLP in 22 eyes (59%). Final mean LogMAR VA was 2.167 ± 0.986 (about counting fingers to hand motions). In the post-cataract endophthalmitis subgroup, 11 eyes (11/27, 41%) achieved favorable visual outcomes with a final mean LogMAR VA of 1.896 ± 1.031 (better than counting fingers). In other subgroups, a total of 10 eyes had no light perception. A total of 19 eyes (51%) of 37 eyes became phthisical. The 3 eyes (8%) of 37 eyes which needed to be eviscerated include 2 post-trabeculectomy eyes and 1 eye with traumatic *E. faecalis* endophthalmitis.

### 3.5. Statistical Analysis

Paired t test showed statistically significant improved visual outcome (*p* = 0.001). Compared with non-cataract subgroups, post-cataract subgroup showed a significant difference of better visual prognosis (*p* = 0.016, chi-squared test). The presenting VA (CF and HM) group significantly achieved a better final visual outcome compared to the group with the worse presenting VA (LP and NLP) (*p* = 0.008, chi-squared test). There was no statistically significant difference between the eyes that underwent PPV and the group without PPV (*p* = 0.919, chi-squared test). This study showed no significant difference in the final visual outcome between the eyes that were treated with and without intravitreal dexamethasone (*p* = 0.692, chi-squared test). Due to the small number of eyes in the etiological subgroups and the non-randomized nature of this retrospective study, statistical conclusions concerning the clinical settings and the intravitreal antibiotics treatment administered were not able to be finalized.

## 4. Discussion

Enterococci are important causative agents of postoperative and posttraumatic endophthalmitis, and enterococcal infections have poor visual prognosis in all causative microorganisms of endophthalmitis. Of the 420 patients enrolled in the Endophthalmitis Vitrectomy Study (EVS), 291 patients had positive culture, with a total of 323 confirmed growth isolates. From these 291 cases, 7 (2.4%) were *Enterococcus* species, including 4 (1.4%) *E. faecalis* [6]. Table 2 illustrates the comparison of published studies and this current study of endophthalmitis caused by *E. faecalis* [5,7,9,10,11,12,13,14,15,16,17,18].

There was no change in the trend of antibiotic susceptibility for *E. faecalis* between our current study and our previous one. *E. faecalis* was equally sensitive to vancomycin, penicillin, ampicillin, and teicoplanin. In terms of resistant to high-level gentamicin, compared with our previous study (8 eyes, 67%), there were six of 22 (27%) in the current series (*p* = 0.026). However, in the post-cataract subgroup, there was no statistical difference between our previous and current study (*p* = 0.116). Resistance to some commonly used antibiotics is a remarkable characteristic of *Enterococcus* species. Although vancomycin-resistant *E. faecalis* is gradually disseminated worldwide, most vancomycin-resistant *E. faecalis* cases are related to infection of other organs or septicemia. In the United States, vancomycin-resistant enterococcal reservoirs include hospital staff and patients, including those that survived hospital stays and residing in skilled nursing facilities; organisms are transmitted by vectors such as stethoscopes, electronic thermometers, sphygmomanometers, and health care workers’ hands [20,21]. There were a few published case reports of vancomycin-resistant enterococci endophthalmitis in Asia [10,16,22]. *E. faecalis* was found to be less commonly resistant to vancomycin. The two enterococcal species reported with higher rate of vancomycin-resistant in endophthalmitis cases were *E.*
*casseliflavus* [23] and *E. faecium* [24]. Meanwhile, intravitreal vancomycin (1 mg/mL) achieves a very high intravitreal concentration of vancomycin against *E. faecalis*, compared with minimal inhibitory concentration of *E. faecalis* in the Clinical and Laboratory Standards Institute standards. Both our previous and current study demonstrated high sensitivity of *E. faecalis* endophthalmitis to vancomycin. Thus, we propose that vancomycin should be continued as the first-line of combined intravitreal antibiotics in the treatment *E. faecalis* endophthalmitis.

Post-cataract surgery, the most common etiology for *E. faecalis* endophthalmitis reported in our study was similarly reported in other studies in the US, Taiwan, South Korea, and Japan [9,14,17,25]. In contrast, the most common etiology reported in majority of the case series in India was trauma-related [10,11,16]. In terms of visual outcome, the post-cataract E. faecalis endophthalmitis patients in this study had a significantly better final VA outcome of better than 20/400 (11 of 27, 41%), when compared to our previous study (1996–2007 case series) (4 of 26, 16%) (*p* = 0.041). The group of patients with a relatively better-presenting VA (CF and HM) achieved a better final visual outcome when compared to the group of patients with presenting VA of LP and NLP (*p* = 0.008). The improvement of final visual outcome was possibly due to early diagnosis with prompt and aggressive treatment of *E. faecalis* endophthalmitis in the post-cataract subgroup.

This study has several limitations. First, the design of this study is retrospective case series. Second, there are a relatively small number of patients in most subgroups, such as traumatic, trabeculectomy-related, and endogenous *E. faecalis* endophthalmitis. This is inevitable given that *E. faecalis* endophthalmitis is not a common infection. This small sample sizes may create inaccuracy in statistical analysis. Third, there are limited groups of antibiotics that were available for sensitivity testing in this tertiary medical institution. However, vancomycin sensitivity response is regularly tested by the laboratory department of our institution. The results are always applicable as vancomycin is routinely indicated as one of the intravitreal or systemic antibiotics in the management of Gram-positive bacterial endophthalmitis in our practice. Fourth, this study does not include non-*E. faecalis* species, such as *E. faecium,* which have been reported to have higher vancomycin resistance rate. This is due to the very small number of non-*E. faecalis* endophthalmitis cases diagnosed during the 12-year period in our institution. Despite all these limitations, this study summarizes a good etiological, antimicrobial sensitivity, management, and outcome profiles of all the consecutive patients diagnosed with *E. faecalis* endophthalmitis in a long 12-year period encountered in our tertiary healthcare institution.

In conclusion, post-cataract surgery was the most common etiology for *E. faecalis* endophthalmitis. All *E. faecalis* isolates were sensitive to vancomycin. Post-cataract *E. faecalis* endophthalmitis had a more favorable final visual prognosis with prompt treatment and intervention. For better prevention and management of endophthalmitis caused by *E. faecalis*, further studies are necessary in evaluating the etiological modification, clinical accuracy, antibiotics strategies, medical and surgical interventions, and most importantly, the functional outcome.

## Figures and Tables

**Table 1 microorganisms-09-00918-t001:** Demographics, Clinical Settings and Features, Managements, and Outcomes of Patients with *Enterococcus faecalis* Endophthalmitis.

Demographics	No. (%)	Clinical Features	No. (%)
Patient number	37	Presenting visual acuity	
Eye affected	37	Counting fingers	1(3%)
OD	17 (46%)	Hand motions	11 (30%)
OS	20 (54%)	Light perception	17 (46%)
Mean age, years	73.0 ± 7.9	No light perception	8 (22%)
Gender		Ocular hypertension	21 (57%)
Male	14 (38%)	Hypopyon	35 (95%)
Female	23 (62%)	Fundus: invisible	37 (100%)
**Systemic illness ***			
Hypertension	13 (35%)	**Management**	
Diabetes	10 (27%)	Initial	
Liver cirrhosis	5 (14%)	Tap	19 (51%)
Cancer	5 (14%)	Vitrectomy	18 (49%)
End-stage renal failure	2 (5%)	Additional	
Coronary arterial disease	2 (5%)	Tap	15 (41%)
Immunocompromised	2 (5%)	Vitrectomy	5 (14%)
Stroke	1 (3%)	Evisceration	3 (8%)
Sepsis	1 (3%)		
Old tuberculosis	1 (3%)	**Final visual acuity**	
**Clinical settings**		≥10/20	1 (3%)
Cataract	27 (73%)	20/50–20/200	8 (22%)
Trabeculectomy	3 (8%)	19/200–20/400	2 (5%)
Endogenous	2 (5%)	4/200-Counting fingers	3 (8%)
Corneal ulcer	2 (5%)	Hand motions	1 (3%)
Vitrectomy	1 (3%)	Light perception	0 (0%)
Pterygium excision	1 (3%)	No light perception	22 (59%)
Trauma	1 (3%)		

*** 7 patients had at least one concomitant systemic illness.

**Table 2 microorganisms-09-00918-t002:** Comparison of studies of endophthalmitis caused by *Enterococcus faecalis.*

	Study	Nationality	Year	No. of	Etiology		Vancomycin		Final VA			
				Eye *		No.	Susceptibility		≥20/400		LP-NLP	
							No.	%	No.	%	No.	%
1	Mao et. al. [7]	US	1977–1990	13/13	NA	13	13/13	100	7/13	54	0	0
2	Booth et al. [18]	US †	1984–1995	28/28	NA		NA		NA †			
3	EVS [5]	US	1990–1994	4/7 ‡	Cataract	4	4/4	100	NA ‡			
4	Scott et al. [17]	US	1990–2001	29/29	Cataract	12	23/23	100	5/29	17	14/29	48
					Trabeculectomy	8						
					PK	4						
					Cataract + trabeculecomy	3						
					Miscellaneous	2						
5	Rishi et al. [10]	India	1995–2007	26/26	Trauma	11	25/26	96	NA **			
					Cataract	7						
					PK	5						
					Endogenous	3						
6	Chen et al. [9]	Taiwan	1996–2007	26/26	Cataract	26	26/26	100	4/26	16	18/26	69
7	Rishi et al. [11]	India	1995–2015	19/19	Trauma	17	19/19	100	12/19	63	6/19	32
	(<18 y/o)				Endogenous	2						
8	Kuriyan et al. [12]	US	2002–2012	14/14	Trabeculectomy	8	13/13	100	1/14	7	9/14	64
					Cataract	4						
					PK	2						
9	Nam et al. [13]	Korea	2004–2010	17/19 ††	Cataract	18	NA	100	5/19 ††	26	7/19 ††	37
					trabeculectomy	1						
10	Todokoro et al. [14]	Japan	NA	9/9	Cataract	9	9/9	100	4/9	44	4/9	44
11	Teng et al. [15]	Taiwan	2004–2015	7/7	Cataract	7	7/7	100	NA ‡‡			
12	Dave et al. [16]	India	2005–2018	9/29 ***	NA	NA	7/9	78	NA ***			
13	Current study	Taiwan	2008–2019	37/37	Cataract	27	37/37	100	11/37	30	22/37	59
					Trabeculectomy	3						
					Endogenous	2						
					Miscellaneous	5						

EVS, endophthalmitis vitrectomy study; LP, light perception; NA, not available; NLP, no light perception; PK, penetrating keratoplasty; VA, visual acuity; US, United States. * Numerator/denominator = number of E. faecalis/total number of Enterococcus species. † Two isolates were form Saudi Arabia. Of 20 cases, 15 had outcomes of 20/200 or worse and 5 had outcomes better than 20/200. ‡ In 7 Enterococcus species, 4 Enterococcus species with VA ≥20/400. ** Final VA ranged from 6/9 to NLP. †† In 19 Enterococcus isolates included 17 E. faecalis and 2 were E. faecium. Final VA was based on 19 Enterococcus species. ‡‡ All eyes with VA worse than 20/100. *** In 29 Enterococcus isolates, 9 were E. faecalis. Final VA was based on 29 Enterococcus species, and 10 were VA ≥ 20/400.

## Data Availability

Not applicable.

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
