# Peer review of "Enterococcus faecalis Endophthalmitis: Clinical Settings, Antibiotic Susceptibility, and Management Outcomes"

_microorganisms, 2021, doi:10.3390/microorganisms9050918_

Round 1
Reviewer 1 Report
The authors present data collected over a 12-year period at their institution on the clinical setting, antibiotic susceptibility, and visual outcomes of E. faecalis endophthalmitis. The data is clearly and concisely presented and the manuscript is well-written. Endophthalmitis is a serious and vision-threatening disease with uniformly poor visual prognoses underlining the importance of studies like the present that help monitor antimicrobial resistance and response to therapies commonly used in practice. The study has limitations as the authors clearly described in the manuscript that goes with the territory of studying an uncommon disease, but the science is sound. I recommend the manuscript for publications with the following minor grammatical corrections. It may be out of the scope of this manuscript, but I would also be interested to know the breakdown of species the other microbial isolates from endophthalmitis cases was during the same 12 year period at authors' institution.
Line 104 – extra period before “Table 1”
Line 155 – I believe there is a grammatical error in this patient that would be fixed if changed to
One patient, “who” displayed complications . . . .
Lines 190 and on - font issues from possible copy and pasting from other documents.
Author Response
The authors present data collected over a 12-year period at their institution on the clinical setting, antibiotic susceptibility, and visual outcomes of E. faecalis endophthalmitis. The data is clearly and concisely presented and the manuscript is well-written. Endophthalmitis is a serious and vision-threatening disease with uniformly poor visual prognoses underlining the importance of studies like the present that help monitor antimicrobial resistance and response to therapies commonly used in practice. The study has limitations as the authors clearly described in the manuscript that goes with the territory of studying an uncommon disease, but the science is sound. I recommend the manuscript for publications with the following minor grammatical corrections. It may be out of the scope of this manuscript, but I would also be interested to know the breakdown of species the other microbial isolates from endophthalmitis cases was during the same 12 year period at authors' institution.
Ans: Thank you for your comments and interest in our manuscript. We add the total number of culture-positive endophthalmitis during the same 12-year period at our institution , and show the percentage of E. faecalis.
During the 12-year period, 457 isolates were cultured from 390 patients with cultured-proven endophthalmitis. Thirty-seven (8.1%) eyes of 37 patients (9.5%) with E. faecalis endophthalmitis were seen during the study period.
Line 104 – extra period before “Table 1”
Ans: Thank for your correction. We delete the extra period.
Line 155 – I believe there is a grammatical error in this patient that would be fixed if changed to One patient, “who” displayed complications . . . .
Ans: Thank for your correction.
One patient, who displayed complications with recurrent endophthalmitis 45 days after the initial infection, managed to achieve a final vision of 20/50 after successfully treated with vitreous tapping and intravenous antibiotics.
Lines 190 and on - font issues from possible copy and pasting from other documents.
Ans: Because this paragraph discusses the virulence factors of E. faecalis, the contents may be not important in this manuscript. We omit this paragraph and avoid plagiarism.

Reviewer 2 Report
This is an interesting report on Enterococcus faecialis endophthalmitis. I have some comments:
- It should be mentioned in the Abstract that this was a retrospective study.
- Introduction: Please refrain from using the term "healthy patients", it should be "healthy subjects" or "otherwise healthy patients", since the term "patient" per se does mean "not healthy".
- Table 1: The authors should also mention, how many of the 37 patients had at least 1 concomitant systemic illness.
- The authors state that 2 cases were immunocompromised, this should also be reflected in Table 1 when describing comorbidities.
- It would be of interest to the readers if the authors could list which antibiotics/treatment regimes were used.
Author Response
- It should be mentioned in the Abstract that this was a retrospective study.
Ans: We edit our abstract.
This retrospective study is to report the clinical settings, antimicrobial susceptibility patterns, and visual outcome of E. faecalis endophthalmitis at a tertiary referral institution in Taoyuan, Taiwan.
- Introduction: Please refrain from using the term "healthy patients", it should be "healthy subjects" or "otherwise healthy patients", since the term "patient" per se does mean "not healthy".
Ans: We follow the comments, and change "healthy patients" to "healthy subjects".
- Table 1: The authors should also mention, how many of the 37 patients had at least 1 concomitant systemic illness.
Ans: We add the number of patients with at least 1 concomitant systemic illness.
- The authors state that 2 cases were immunocompromised, this should also be reflected in Table 1 when describing comorbidities.
Ans: We state that 2 cases were immunocompromised in Table 1.
- It would be of interest to the readers if the authors could list which antibiotics/treatment regimes were used.
Ans: We add the antibiotics/treatment regimes in method section.
Before S. pneumoniae was cultured, intravitreal antibiotics included vancomycin with either ceftazidime or amikacin. After positive cultures of S. pneumoniae isolates and antibiotic susceptibility testing results were obtained, intravitreal vancomycin was administered.

Round 2
Reviewer 2 Report
I have no further comments.